# Learning to Pivot with Adversarial Networks

**Gilles Louppe**
New York University
g.louppe@nyu.edu

**Michael Kagan**
SLAC National Accelerator Laboratory
makagan@slac.stanford.edu

**Kyle Cranmer**
New York University
kyle.cranmer@nyu.edu

## Abstract

Several techniques for domain adaptation have been proposed to account for differences in the distribution of the data used for training and testing. The majority of this work focuses on a binary domain label. Similar problems occur in a scientific context where there may be a continuous family of plausible data generation processes associated to the presence of systematic uncertainties. Robust inference is possible if it is based on a pivot – a quantity whose distribution does not depend on the unknown values of the nuisance parameters that parametrize this family of data generation processes. In this work, we introduce and derive theoretical results for a training procedure based on adversarial networks for enforcing the pivotal property (or, equivalently, fairness with respect to continuous attributes) on a predictive model. The method includes a hyperparameter to control the trade-off between accuracy and robustness. We demonstrate the effectiveness of this approach with a toy example and examples from particle physics.

## 1   Introduction

Machine learning techniques have been used to enhance a number of scientific disciplines, and they have the potential to transform even more of the scientific process. One of the challenges of applying machine learning to scientific problems is the need to incorporate systematic uncertainties, which affect both the robustness of inference and the metrics used to evaluate a particular analysis strategy.

In this work, we focus on supervised learning techniques where systematic uncertainties can be associated to a data generation process that is not uniquely specified. In other words, the lack of systematic uncertainties corresponds to the (rare) case that the process that generates training data is unique, fully specified, and an accurate representative of the real world data. By contrast, a common situation when systematic uncertainty is present is when the training data are not representative of the real data. Several techniques for domain adaptation have been developed to create models that are more robust to this binary type of uncertainty. A more generic situation is that there are several plausible data generation processes, specified as a family parametrized by continuous nuisance parameters, as is typically found in scientific domains. In this broader context, statisticians have for long been working on robust inference techniques based on the concept of a pivot – a quantity whose distribution is invariant with the nuisance parameters (see e.g., (Degroot and Schervish, 1975)).

Assuming a probability model $p(X, Y, Z)$, where $X$ are the data, $Y$ are the target labels, and $Z$ are the nuisance parameters, we consider the problem of learning a predictive model $f(X)$ for $Y$ conditional on the observed values of $X$ that is robust to uncertainty in the unknown value of $Z$. We introduce a flexible learning procedure based on adversarial networks (Goodfellow et al., 2014) for enforcing that $f(X)$ is a pivot with respect to $Z$. We derive theoretical results proving that the procedure converges towards a model that is both optimal and statistically independent of the nuisance parameters (if that model exists) or for which one can tune a trade-off between accuracy and robustness (e.g., as driven by a higher level objective). In particular, and to the best of our knowledge, our contribution is the first solution for imposing pivotal constraints on a predictive model, working regardless of the

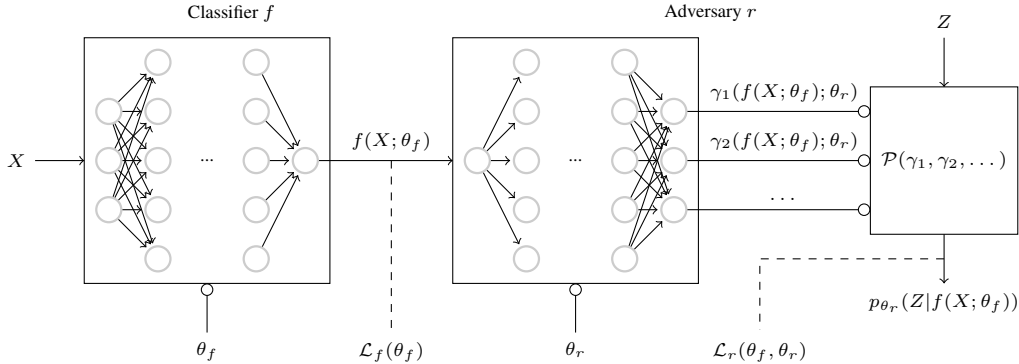

Figure 1: Architecture for the adversarial training of a binary classifier $f$ against a nuisance parameters $Z$. The adversary $r$ models the distribution $p(z|f(X;\theta_f) = s)$ of the nuisance parameters as observed only through the output $f(X;\theta_f)$ of the classifier. By maximizing the antagonistic objective $\mathcal{L}_r(\theta_f, \theta_r)$, the classifier $f$ forces $p(z|f(X;\theta_f) = s)$ towards the prior $p(z)$, which happens when $f(X;\theta_f)$ is independent of the nuisance parameter $Z$ and therefore pivotal.

type of the nuisance parameter (discrete or continuous) or of its prior. Finally, we demonstrate the effectiveness of the approach with a toy example and examples from particle physics.

## 2 Problem statement

We begin with a family of data generation processes $p(X, Y, Z)$, where $X \in \mathcal{X}$ are the data, $Y \in \mathcal{Y}$ are the target labels, and $Z \in \mathcal{Z}$ are the nuisance parameters that can be continuous or categorical. Let us assume that prior to incorporating the effect of uncertainty in $Z$, our goal is to learn a regression function $f : \mathcal{X} \to \mathcal{S}$ with parameters $\theta_f$ (e.g., a neural network-based probabilistic classifier) that minimizes a loss $\mathcal{L}_f(\theta_f)$ (e.g., the cross-entropy). In classification, values $s \in \mathcal{S} = \mathbb{R}^{|\mathcal{Y}|}$ correspond to the classifier scores used for mapping hard predictions $y \in \mathcal{Y}$, while $\mathcal{S} = \mathcal{Y}$ for regression.

We augment our initial objective so that inference based on $f(X;\theta_f)$ will be robust to the value $z \in \mathcal{Z}$ of the nuisance parameter $Z$ – which remains unknown at test time. A formal way of enforcing robustness is to require that the distribution of $f(X;\theta_f)$ conditional on $Z$ (and possibly $Y$) be invariant with the nuisance parameter $Z$. Thus, we wish to find a function $f$ such that

$$p(f(X;\theta_f) = s|z) = p(f(X;\theta_f) = s|z') \tag{1}$$

for all $z, z' \in \mathcal{Z}$ and all values $s \in \mathcal{S}$ of $f(X;\theta_f)$. In words, we are looking for a predictive function $f$ which is a pivotal quantity with respect to the nuisance parameters. This implies that $f(X;\theta_f)$ and $Z$ are independent random variables.

As stated in Eqn. 1, the pivotal quantity criterion is imposed with respect to $p(X|Z)$ where $Y$ is marginalized out. In some situations however (see e.g., Sec. 5.2), class conditional independence of $f(X;\theta_f)$ on the nuisance $Z$ is preferred, which can then be stated as requiring

$$p(f(X;\theta_f) = s|z, y) = p(f(X;\theta_f) = s|z', y) \tag{2}$$

for one or several specified values $y \in \mathcal{Y}$.

## 3 Method

Joint training of adversarial networks was first proposed by (Goodfellow et al., 2014) as a way to build a generative model capable of producing samples from random noise $z$. More specifically, the authors pit a generative model $g : \mathbb{R}^n \to \mathbb{R}^p$ against an adversarial classifier $d : \mathbb{R}^p \to [0, 1]$ whose antagonistic objective is to recognize real data $X$ from generated data $g(Z)$. Both models $g$ and $d$ are trained simultaneously, in such a way that $g$ learns to produce samples that are difficult to identify by $d$, while $d$ incrementally adapts to changes in $g$. At the equilibrium, $g$ models a distribution whose samples can be identified by $d$ only by chance. That is, assuming enough capacity in $d$ and $g$, the distribution of $g(Z)$ eventually converges towards the real distribution of $X$.

---

**Algorithm 1** Adversarial training of a classifier $f$ against an adversary $r$.

---

*Inputs:* training data $\{x_i, y_i, z_i\}_{i=1}^N$; *Outputs:* $\hat{\theta}_f, \hat{\theta}_r$.

1: **for** $t = 1$ to $T$ **do**
2:     **for** $k = 1$ to $K$ **do**
3:         Sample minibatch $\{x_m, z_m, s_m = f(x_m; \theta_f)\}_{m=1}^M$ of size $M$;
4:         With $\theta_f$ fixed, update $r$ by ascending its stochastic gradient $\nabla_{\theta_r} E(\theta_f, \theta_r) :=$

$$\nabla_{\theta_r} \sum_{m=1}^M \log p_{\theta_r}(z_m | s_m);$$

5:     **end for**
6:     Sample minibatch $\{x_m, y_m, z_m, s_m = f(x_m; \theta_f)\}_{m=1}^M$ of size $M$;
7:     With $\theta_r$ fixed, update $f$ by descending its stochastic gradient $\nabla_{\theta_f} E(\theta_f, \theta_r) :=$

$$\nabla_{\theta_f} \sum_{m=1}^M \left[ -\log p_{\theta_f}(y_m | x_m) + \log p_{\theta_r}(z_m | s_m) \right],$$

    where $p_{\theta_f}(y_m | x_m)$ denotes $\mathbf{1}(y_m = 0)(1 - s_m) + \mathbf{1}(y_m = 1)s_m$;
8: **end for**

---

In this work, we repurpose adversarial networks as a means to constrain the predictive model $f$ in order to satisfy Eqn. 1. As illustrated in Fig. 1, we pit $f$ against an adversarial model $r :=$ $p_{\theta_r}(z | f(X; \theta_f) = s)$ with parameters $\theta_r$ and associated loss $\mathcal{L}_r(\theta_f, \theta_r)$. This model takes as input realizations $s$ of $f(X; \theta_f)$ and produces as output a function modeling the posterior probability density $p_{\theta_r}(z | f(X; \theta_f) = s)$. Intuitively, if $p(f(X; \theta_f) = s | z)$ varies with $z$, then the corresponding correlation can be captured by $r$. By contrast, if $p(f(X; \theta_f) = s | z)$ is invariant with $z$, as we require, then $r$ should perform poorly and be close to random guessing. Training $f$ such that it additionally minimizes the performance of $r$ therefore acts as a regularization towards Eqn. 1.

If $Z$ takes discrete values, then $p_{\theta_r}$ can be represented as a probabilistic classifier $\mathbb{R} \to \mathbb{R}^{|\mathcal{Z}|}$ whose $j^{\text{th}}$ output (for $j = 1, \ldots, |\mathcal{Z}|$) is the estimated probability mass $p_{\theta_r}(z_j | f(X; \theta_f) = s)$. Similarly, if $Z$ takes continuous values, then we can model the posterior probability density $p(z | f(X; \theta_f) = s)$ with a sufficiently flexible parametric family of distributions $\mathcal{P}(\gamma_1, \gamma_2, \dots)$, where the parameters $\gamma_j$ depend on $f(X, \theta_f)$ and $\theta_r$. The adversary $r$ may take any form, i.e. it does not need to be a neural network, as long as it exposes a differentiable function $p_{\theta_r}(z | f(X; \theta_f) = s)$ of sufficient capacity to represent the true distribution. Fig. 1 illustrates a concrete example where $p_{\theta_r}(z | f(X; \theta_f) = s)$ is a mixture of gaussians, as modeled with a mixture density network (Bishop, 1994)). The $j^{\text{th}}$ output corresponds to the estimated value of the corresponding parameter $\gamma_j$ of that distribution (e.g., the mean, variance and mixing coefficients of its components). The estimated probability density $p_{\theta_r}(z | f(X; \theta_f) = s)$ can then be evaluated for any $z \in \mathcal{Z}$ and any score $s \in \mathcal{S}$.

As with generative adversarial networks, we propose to train $f$ and $r$ simultaneously, which we carry out by considering the value function

$$E(\theta_f, \theta_r) = \mathcal{L}_f(\theta_f) - \mathcal{L}_r(\theta_f, \theta_r) \tag{3}$$

that we optimize by finding the minimax solution

$$\hat{\theta}_f, \hat{\theta}_r = \arg\min_{\theta_f} \max_{\theta_r} E(\theta_f, \theta_r). \tag{4}$$

Without loss of generality, the adversarial training procedure to obtain $(\hat{\theta}_f, \hat{\theta}_r)$ is formally presented in Algorithm 1 in the case of a binary classifier $f : \mathbb{R}^p \to [0, 1]$ modeling $p(Y = 1 | X)$. For reasons further explained in Sec. 4, $\mathcal{L}_f$ and $\mathcal{L}_r$ are respectively set to the expected value of the negative log-likelihood of $Y | X$ under $f$ and of $Z | f(X; \theta_f)$ under $r$:

$$\mathcal{L}_f(\theta_f) = \mathbb{E}_{x \sim X} \mathbb{E}_{y \sim Y | x}[-\log p_{\theta_f}(y | x)], \tag{5}$$

$$\mathcal{L}_r(\theta_f, \theta_r) = \mathbb{E}_{s \sim f(X; \theta_f)} \mathbb{E}_{z \sim Z | s}[-\log p_{\theta_r}(z | s)]. \tag{6}$$

The optimization algorithm consists in using stochastic gradient descent alternatively for solving Eqn. 4. Finally, in the case of a class conditional pivot, the settings are the same, except that the adversarial term $\mathcal{L}_r(\theta_f, \theta_r)$ is restricted to $Y = y$.

# 4 Theoretical results

In this section, we show that in the setting of Algorithm 1 where $\mathcal{L}_f$ and $\mathcal{L}_r$ are respectively set to expected value of the negative log-likelihood of $Y|X$ under $f$ and of $Z|f(X;\theta_f)$ under $r$, the minimax solution of Eqn. 4 corresponds to a classifier $f$ which is a pivotal quantity.

In this setting, the nuisance parameter $Z$ is considered as a random variable of prior $p(Z)$, and our goal is to find a function $f(\cdot;\theta_f)$ such that $f(X;\theta_f)$ and $Z$ are independent random variables. Importantly, classification of $Y$ with respect to $X$ is considered in the context where $Z$ is marginalized out, which means that the classifier minimizing $\mathcal{L}_f$ is optimal with respect to $Y|X$, but not necessarily with $Y|X, Z$. Results hold for a nuisance parameters $Z$ taking either categorical or continuous values. By abuse of notation, $H(Z)$ denotes the differential entropy in this latter case. Finally, the proposition below is derived in a non-parametric setting, by assuming that both $f$ and $r$ have enough capacity.

**Proposition 1.** *If there exists a minimax solution $(\hat{\theta}_f, \hat{\theta}_r)$ for Eqn. 4 such that $E(\hat{\theta}_f, \hat{\theta}_r) = H(Y|X) - H(Z)$, then $f(\cdot;\hat{\theta}_f)$ is both an optimal classifier and a pivotal quantity.*

*Proof.* For fixed $\theta_f$, the adversary $r$ is optimal at

$$\hat{\hat{\theta}}_r = \arg\max_{\theta_r} E(\theta_f, \theta_r) = \arg\min_{\theta_r} \mathcal{L}_r(\theta_f, \theta_r), \tag{7}$$

in which case $p_{\hat{\hat{\theta}}_r}(z|f(X;\theta_f) = s) = p(z|f(X;\theta_f) = s)$ for all $z$ and all $s$, and $\mathcal{L}_r$ reduces to the expected entropy $\mathbb{E}_{s\sim f(X;\theta_f)}[H(Z|f(X;\theta_f) = s)]$ of the conditional distribution of the nuisance parameters. This expectation corresponds to the conditional entropy of the random variables $Z$ and $f(X;\theta_f)$ and can be written as $H(Z|f(X;\theta_f))$. Accordingly, the value function $E$ can be restated as a function depending on $\theta_f$ only:

$$E'(\theta_f) = \mathcal{L}_f(\theta_f) - H(Z|f(X;\theta_f)). \tag{8}$$

In particular, we have the lower bound

$$H(Y|X) - H(Z) \leq \mathcal{L}_f(\theta_f) - H(Z|f(X;\theta_f)) \tag{9}$$

where the equality holds at $\hat{\theta}_f = \arg\min_{\theta_f} E'(\theta_f)$ when:

- $\hat{\theta}_f$ minimizes the negative log-likelihood of $Y|X$ under $f$, which happens when $\hat{\theta}_f$ are the parameters of an optimal classifier. In this case, $\mathcal{L}_f$ reduces to its minimum value $H(Y|X)$.

- $\hat{\theta}_f$ maximizes the conditional entropy $H(Z|f(X;\theta_f))$, since $H(Z|f(X;\theta)) \leq H(Z)$ from the properties of entropy. Note that this latter inequality holds for both the discrete and the differential definitions of entropy.

By assumption, the lower bound is active, thus we have $H(Z|f(X;\theta_f)) = H(Z)$ because of the second condition, which happens exactly when $Z$ and $f(X;\theta_f)$ are independent variables. In other words, the optimal classifier $f(\cdot;\hat{\theta}_f)$ is also a pivotal quantity. $\square$

Proposition 1 suggests that if at each step of Algorithm 1 the adversary $r$ is allowed to reach its optimum given $f$ (e.g., by setting $K$ sufficiently high) and if $f$ is updated to improve $\mathcal{L}_f(\theta_f) - H(Z|f(X;\theta_f))$ with sufficiently small steps, then $f$ should converge to a classifier that is both optimal and pivotal, provided such a classifier exists. Therefore, the adversarial term $\mathcal{L}_r$ can be regarded as a way to select among the class of all optimal classifiers a function $f$ that is also pivotal. Despite the former theoretical characterization of the minimax solution of Eqn. 4, let us note that formal guarantees of convergence towards that solution by Algorithm 1 in the case where a finite number $K$ of steps is taken for $r$ remains to be proven.

In practice, the assumption of existence of an optimal and pivotal classifier may not hold because the nuisance parameter directly shapes the decision boundary. In this case, the lower bound

$$H(Y|X) - H(Z) < \mathcal{L}_f(\theta_f) - H(Z|f(X;\theta_f)) \tag{10}$$

is strict: $f$ can either be an optimal classifier or a pivotal quantity, but not both simultaneously. In this situation, it is natural to rewrite the value function $E$ as

$$E_\lambda(\theta_f, \theta_r) = \mathcal{L}_f(\theta_f) - \lambda\mathcal{L}_r(\theta_f, \theta_r), \tag{11}$$

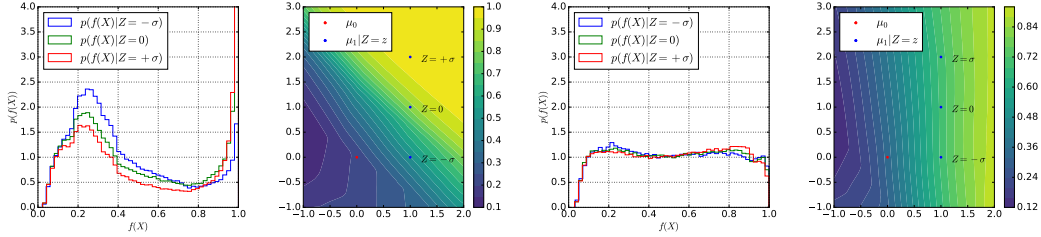

Figure 2: Toy example. (Left) Conditional probability densities of the decision scores at $Z = -\sigma, 0, \sigma$ without adversarial training. The resulting densities are dependent on the continuous parameter $Z$, indicating that $f$ is not pivotal. (Middle left) The associated decision surface, highlighting the fact that samples are easier to classify for values of $Z$ above $\sigma$, hence explaining the dependency. (Middle right) Conditional probability densities of the decision scores at $Z = -\sigma, 0, \sigma$ when $f$ is built with adversarial training. The resulting densities are now almost identical to each other, indicating only a small dependency on $Z$. (Right) The associated decision surface, illustrating how adversarial training bends the decision function vertically to erase the dependency on $Z$.

where $\lambda \geq 0$ is a hyper-parameter controlling the trade-off between the performance of $f$ and its independence with respect to the nuisance parameter. Setting $\lambda$ to a large value will preferably enforces $f$ to be pivotal while setting $\lambda$ close to 0 will rather constraint $f$ to be optimal. When the lower bound is strict, let us note however that there may exist distinct but equally good solutions $\theta_f, \theta_r$ minimizing Eqn. 11. In this zero-sum game, an increase in accuracy would exactly be compensated by a decrease in pivotality and vice-versa. How to best navigate this Pareto frontier to maximize a higher-level objective remains a question open for future works.

Interestingly, let us finally emphasize that our results hold using only the (1D) output $s$ of $f(\cdot; \theta_f)$ as input to the adversary. We could similarly enforce an intermediate representation of the data to be pivotal, e.g. as in (Ganin and Lempitsky, 2014), but this is not necessary.

## 5   Experiments

In this section, we empirically demonstrate the effectiveness of the approach with a toy example and examples from particle physics. Notably, there are no other other approaches to compare to in the case of continuous nuisance parameters, as further explained in Sec. 6. In the case of binary parameters, we do not expect results to be much different from previous works. The source code to reproduce the experiments is available online [1].

### 5.1   A toy example with a continous nuisance parameter

As a guiding toy example, let us consider the binary classification of 2D data drawn from multivariate gaussians with equal priors, such that

$$x \sim \mathcal{N}\left((0,0), \begin{bmatrix} 1 & -0.5 \\ -0.5 & 1 \end{bmatrix}\right) \qquad \text{when } Y = 0, \qquad (12)$$

$$x|Z=z \sim \mathcal{N}\left((1, 1+z), \begin{bmatrix} 1 & 0 \\ 0 & 1 \end{bmatrix}\right) \qquad \text{when } Y = 1. \qquad (13)$$

The continuous nuisance parameter $Z$ here represents our uncertainty about the location of the mean of the second gaussian. Our goal is to build a classifier $f(\cdot; \theta_f)$ for predicting $Y$ given $X$, but such that the probability distribution of $f(X; \theta_f)$ is invariant with respect to the nuisance parameter $Z$.

Assuming a gaussian prior $z \sim \mathcal{N}(0, 1)$, we generate data $\{x_i, y_i, z_i\}_{i=1}^N$, from which we train a neural network $f$ minimizing $\mathcal{L}_f(\theta_f)$ without considering its adversary $r$. The network architecture comprises 2 dense hidden layers of 20 nodes respectively with tanh and ReLU activations, followed by a dense output layer with a single node with a sigmoid activation. As shown in Fig. 2, the resulting classifier is not pivotal, as the conditional probability densities of its decision scores $f(X; \theta_f)$ show

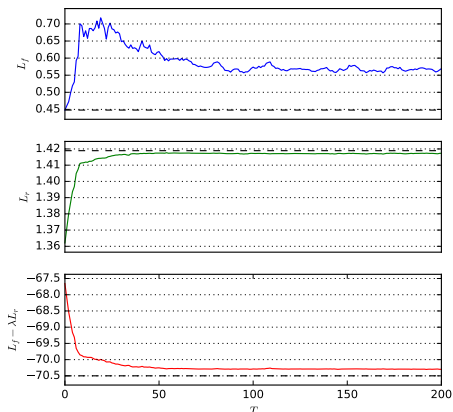

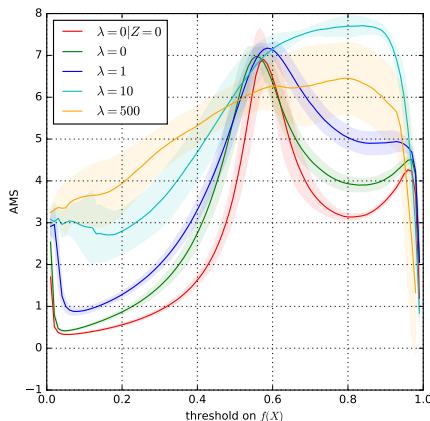

Figure 3: Toy example. Training curves for $\mathcal{L}_f(\theta_f)$, $\mathcal{L}_r(\theta_f, \theta_r)$ and $\mathcal{L}_f(\theta_f) - \lambda\mathcal{L}_r(\theta_f, \theta_r)$. Initialized with a pre-trained classifier $f$, adversarial training was performed for 200 iterations, mini-batches of size $M = 128$, $K = 500$ and $\lambda = 50$.

Figure 4: Physics example. Approximate median significance as a function of the decision threshold on the output of $f$. At $\lambda = 10$, trading accuracy for independence to pileup results in a net benefit in terms of statistical significance.

large discrepancies between values $z$ of the nuisance parameters. While not shown here, a classifier trained only from data generated at the nominal value $Z = 0$ would also not be pivotal.

Let us now consider the joint training of $f$ against an adversary $r$ implemented as a mixture density network modeling $Z|f(X; \theta_f)$ as a mixture of five gaussians. The network architecture of $r$ comprises 2 dense hidden layers of 20 nodes with ReLU activations, followed by an output layer of 15 nodes corresponding to the means, standard deviations and mixture coefficients of the gaussians. Output nodes for the mean values come with linear activations, output nodes for the standard deviations with exponential activations to ensure positivity, while output nodes for the mixture coefficients implement the softmax function to ensure positivity and normalization. When running Algorithm 1 as initialized with the classifier $f$ obtained previously, adversarial training effectively reshapes the decision function so it that becomes almost independent on the nuisance parameter, as shown in Fig. 2. The conditional probability densities of the decision scores $f(X; \theta_f)$ are now very similar to each other, indicating only a residual dependency on the nuisance, as theoretically expected. The dynamics of adversarial training is illustrated in Fig. 3, where the losses $\mathcal{L}_f$, $\mathcal{L}_r$ and $\mathcal{L}_f - \lambda\mathcal{L}_r$ are evaluated after each iteration. In the first iterations, we observe that the global objective $\mathcal{L}_f - \lambda\mathcal{L}_r$ is minimized by making the classifier less accurate, hence the corresponding increase of $\mathcal{L}_f$, but which results in a classifier that is more pivotal, hence the associated increase of $\mathcal{L}_r$ and the total net benefit. As learning goes, minimizing $E$ requires making predictions that are more accurate, hence decreasing $\mathcal{L}_f$, or that are even less dependent on $Z$, hence shaping $p_{\theta_r}$ towards the prior $p(Z)$. Indeed, $\mathcal{L}_f$ eventually starts decreasing, while remaining bounded from below by $\min_{\theta_f} \mathcal{L}_f(\theta_f)$ as approximated by the dashed line in the first plot. Similarly, $\mathcal{L}_r$ tends towards the differential entropy $H(Z)$ of the prior (where $H(Z) = \log(\sigma\sqrt{2\pi e}) = 1.419$ in the case of a standard normal), as shown by the dashed line in the second plot. Finally, let us note that the ideal situation of a classifier that is both optimal and pivotal is unreachable for this problem, as shown in the third plot by the offset between $\mathcal{L}_f - \lambda\mathcal{L}_r$ and the dashed line approximating $H(Y|X) - \lambda H(Z)$.

## 5.2 High energy physics examples

**Binary Case** Experiments at high energy colliders like the LHC (Evans and Bryant, 2008) are searching for evidence of new particles beyond those described by the Standard Model (SM) of particle physics. A wide array of theories predict the existence of new massive particles that would decay to known particles in the SM such as the $W$ boson. The $W$ boson is unstable and can decay to two quarks, each of which produce collimated sprays of particles known as jets. If the exotic particle is heavy, then the $W$ boson will be moving very fast, and relativistic effects will cause the two jets from its decay to merge into a single '$W$-jet'. These $W$-jets have a rich internal substructure. However, jets are also produced ubiquitously at high energy colliders through more mundane processes in the

SM, which leads to a challenging classification problem that is beset with a number of sources of systematic uncertainty. The classification challenge used here is common in jet substructure studies (see e.g. (CMS Collaboration, 2014; ATLAS Collaboration, 2015, 2014)): we aim to distinguish normal jets produced copiously at the LHC ($Y = 0$) and from $W$-jets ($Y = 1$) potentially coming from an exotic process. We reuse the datasets used in (Baldi et al., 2016a).

Challenging in its own right, this classification problem is made all the more difficult by the presence of pileup, or multiple proton-proton interactions occurring simultaneously with the primary interaction. These pileup interactions produce additional particles that can contribute significant energies to jets unrelated to the underlying discriminating information. The number of pileup interactions can vary with the running conditions of the collider, and we want the classifier to be robust to these conditions. Taking some liberty, we consider an extreme case with a categorical nuisance parameter, where $Z = 0$ corresponds to events without pileup and $Z = 1$ corresponds to events with pileup, for which there are an average of 50 independent pileup interactions overlaid.

We do not expect that we will be able to find a function $f$ that simultaneously minimizes the classification loss $\mathcal{L}_f$ and is pivotal. Thus, we need to optimize the hyper-parameter $\lambda$ of Eqn. 11 with respect to a higher-level objective. In this case, the natural higher-level context is a hypothesis test of a null hypothesis with no $Y = 1$ events against an alternate hypothesis that is a mixture of $Y = 0$ and $Y = 1$ events. In the absence of systematic uncertainties, optimizing $\mathcal{L}_f$ simultaneously optimizes the power of a classical hypothesis test in the Neyman-Pearson sense. When we include systematic uncertainties we need to balance the classification performance against the robustness to uncertainty in $Z$. Since we are still performing a hypothesis test against the null, we only wish to impose the pivotal property on $Y = 0$ events. To this end, we use as a higher level objective the Approximate Median Significance (AMS), which is a natural generalization of the power of a hypothesis test when systematic uncertainties are taken into account (see Eqn. 20 of Adam-Bourdarios et al. (2014)).

For several values of $\lambda$, we train a classifier using Algorithm 1 but consider the adversarial term $\mathcal{L}_r$ conditioned on $Y = 0$ only, as outlined in Sec. 2. The architecture of $f$ comprises 3 hidden layers of 64 nodes respectively with tanh, ReLU and ReLU activations, and is terminated by a single final output node with a sigmoid activation. The architecture of $r$ is the same, but uses only ReLU activations in its hidden nodes. As in the previous example, adversarial training is initialized with $f$ pre-trained. Experiments are performed on a subset of 150000 samples for training while AMS is evaluated on an independent test set of 5000000 samples. Both training and testing samples are weighted such that the null hypothesis corresponded to 1000 of $Y = 0$ events and the alternate hypothesis included an additional 100 $Y = 1$ events prior to any thresholding on $f$. This allows us to probe the efficacy of the method proposed here in a representative background-dominated high energy physics environment. Results reported below are averages over 5 runs.

As Fig. 4 illustrates, without adversarial training (at $\lambda = 0|Z = 0$ when building a classifier at the nominal value $Z = 0$ only, or at $\lambda = 0$ when building a classifier on data sampled from $p(X, Y, Z)$), the AMS peaks at 7. By contrast, as the pivotal constraint is made stronger (for $\lambda > 0$) the AMS peak moves higher, with a maximum value around 7.8 for $\lambda = 10$. Trading classification accuracy for robustness to pileup thereby results in a net benefit in terms of the power of the hypothesis test. Setting $\lambda$ too high however (e.g. $\lambda = 500$) results in a decrease of the maximum AMS, by focusing the capacity of $f$ too strongly on independence with $Z$, at the expense of accuracy. In effect, optimizing $\lambda$ yields a principled and effective approach to control the trade-off between accuracy and robustness that ultimately maximizes the power of the enveloping hypothesis test.

**Continous Case** Recently, an independent group has used our approach to learn jet classifiers that are independent of the jet mass (Shimmin et al., 2017), which is a continuous attribute. The results of their studies show that the adversarial training strategy works very well for real-world problems with continuous attributes, thus enhancing the sensitivity of searches for new physics at the LHC.

# 6 Related work

Learning to pivot can be related to the problem of domain adaptation (Blitzer et al., 2006; Pan et al., 2011; Gopalan et al., 2011; Gong et al., 2013; Baktashmotlagh et al., 2013; Ajakan et al., 2014; Ganin and Lempitsky, 2014), where the goal is often stated as trying to learn a domain-invariant representation of the data. Likewise, our method also relates to the problem of enforcing fairness

in classification (Kamishima et al., 2012; Zemel et al., 2013; Feldman et al., 2015; Edwards and Storkey, 2015; Zafar et al., 2015; Louizos et al., 2015), which is stated as learning a classifier that is independent of some chosen attribute such as gender, color or age. For both families of methods, the problem can equivalently be stated as learning a classifier which is a pivotal quantity with respect to either the domain or the selected feature. As an example, unsupervised domain adaptation with labeled data from a source domain and unlabeled data from a target domain can be recast as learning a predictive model $f$ (i.e., trained to minimize $\mathcal{L}_f$ evaluated on labeled source data only) that is also a pivot with respect to the domain $Z$ (i.e., trained to maximize $\mathcal{L}_r$ evaluated on both source and target data). In this context, (Ganin and Lempitsky, 2014; Edwards and Storkey, 2015) are certainly among the closest to our work, in which domain invariance and fairness are enforced through an adversarial minimax setup composed of a classifier and an adversarial discriminator. Following this line of work, our method can be regarded as a unified generalization that also supports a continuously parametrized family of domains or as enforcing fairness over continuous attributes.

Most related work is based on the strong and limiting assumption that $Z$ is a binary random variable (e.g., $Z = 0$ for the source domain, and $Z = 1$ for the target domain). In particular, (Pan et al., 2011; Gong et al., 2013; Baktashmotlagh et al., 2013; Zemel et al., 2013; Ganin and Lempitsky, 2014; Ajakan et al., 2014; Edwards and Storkey, 2015; Louizos et al., 2015) are all based on the minimization of some form of divergence between the two distributions of $f(X)|Z = 0$ and $f(X)|Z = 1$. For this reason, these works cannot directly be generalized to non-binary or continuous nuisance parameters, both from a practical and theoretical point of view. Notably, Kamishima et al. (2012) enforces fairness through a prejudice regularization term based on empirical estimates of $p(f(X)|Z)$. While this approach is in principle sufficient for handling non-binary nuisance parameters $Z$, it requires accurate empirical estimates of $p(f(X)|Z = z)$ for all values $z$, which quickly becomes impractical as the cardinality of $Z$ increases. By contrast, our approach models the conditional dependence through an adversarial network, which allows for generalization without necessarily requiring an exponentially growing number of training examples.

A common approach to account for systematic uncertainties in a scientific context (e.g. in high energy physics) is to take as fixed a classifier $f$ built from training data for a nominal value $z_0$ of the nuisance parameter, and then propagate uncertainty by estimating $p(f(x)|z)$ with a parametrized calibration procedure. Clearly, this classifier is however not optimal for $z \neq z_0$. To overcome this issue, the classifier $f$ is sometimes built instead on a mixture of training data generated from several plausible values $z_0, z_1, \ldots$ of the nuisance parameter. While this certainly improves classification performance with respect to the marginal model $p(X, Y)$, there is no reason to expect the resulting classifier to be pivotal, as shown previously in Sec. 5.1. As an alternative, parametrized classifiers (Cranmer et al., 2015; Baldi et al., 2016b) directly take (nuisance) parameters as additional input variables, hence ultimately providing the most statistically powerful approach for incorporating the effect of systematics on the underlying classification task. In practice, parametrized classifiers are also computationally expensive to build and evaluate. In particular, calibrating their decision function, i.e. approximating $p(f(x, z)|y, z)$ as a continuous function of $z$, remains an open challenge. By contrast, constraining $f$ to be pivotal yields a classifier that can be directly used in a wider range of applications, since the dependence on the nuisance parameter $Z$ has already been eliminated.

## 7 Conclusions

In this work, we proposed a flexible learning procedure for building a predictive model that is independent of continuous or categorical nuisance parameters by jointly training two neural networks in an adversarial fashion. From a theoretical perspective, we motivated the proposed algorithm by showing that the minimax value of its value function corresponds to a predictive model that is both optimal and pivotal (if that models exists) or for which one can tune the trade-off between power and robustness. From an empirical point of view, we confirmed the effectiveness of our method on a toy example and a particle physics example.

In terms of applications, our solution can be used in any situation where the training data may not be representative of the real data the predictive model will be applied to in practice. In the scientific context, the presence of systematic uncertainty can be incorporated by considering a family of data generation processes, and it would be worth revisiting those scientific problems that utilize machine learning in light of this technique. The approach also extends to cases where independence of the predictive model with respect to observed random variables is desired, as in fairness for classification.

## Acknowledgements

We would like to thank the authors of (Baldi et al., 2016a) for sharing the data used in their studies. KC and GL are both supported through NSF ACI-1450310, additionally KC is supported through PHY-1505463 and PHY-1205376. MK is supported by the US Department of Energy (DOE) under grant DE-AC02-76SF00515 and by the SLAC Panofsky Fellowship.

## Footnotes

[1]`https://github.com/glouppe/paper-learning-to-pivot`

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
