[Reviews · NeurIPS 2017]

Reviewer 1



The paper proposes an adversarial framework for training a classifier that is robust to systematic uncertainties. The paper studies a scenario where the data generation process of interest is governed by an unknown random variable Z where the random variable defines a family of data generation distribution (i.e. a different distribution is incurred by a realization of the random variable). It then applies the proposed method to train a classifier whose performance is independent of the random variable. The algorithm was tested on a synthetic dataset and a high energy physics example. No comparison with prior arts were drawn. Paper strength The paper extends the generative adversarial framework to a robust classification problem against systematic noise in the data generation mechanism. The paper shows that the optimal classifier of the proposed framework is a classifier whose output is independent to the input variable. Paper weakness The paper fails to motivate the problem. Due to the difference of the members in the family of data distribution, samples from some member is naturally easier to classify than members from the other distribution (For example, consider the case that z is very large in Section 5.1). In other words, it is common to have classification outputs correlates with the unknown variable. A classifier would do better for samples from some members and do worse for samples from the other members. By adding the proposed adversary, which encourages the output to be independent with Z. It basically forces the classifier to do worse for the easy cases for matching the performance for the hard cases. The experiment results reports in the paper also suggest this is what happens. The overall accuracy is traded-off for consistency. It is unclear why it is advantageous to do so. A more reasonable approach would be to improve the performance across the members. No comparison with prior works. The paper does not report comparison with baseline approaches that also consider the existence of Z when training the classifier. The only baseline model is a plane classifier model that ignores the existence of Z. Experiment details are missing. Many details such as the data format in the section 5.2 is missing. It is difficult to understand the experiment setting. The discussions on whether the optimal and pivotal classifier is achievable is very vague. The results that the training curves shown in the paper are away from the optimal values do not mean that the optimal and pivotal classifier is not achievable. The achievability might be just limited by the training of the network capacity. Hence, the statement is not solid.

Reviewer 2



This paper addresses accounting for unknown nuisance parameters that may have introduced selection bias and/or confound classification results. The paper is very interesting, and I feel the approach is important and significant in a number of settings. Unfortunately, I feel like the experiments are rather limited. Particularly, the authors make the claim that there are no datasets with continuous pivot variables to train on, but this can be easily constructed. For instance, one could select MNIST (or CIFAR, Imagenet, etc) images with a selection based on the intensity or variance over the pixels (which could be modeled a number of ways). In addition, this should be applicable to the base where there is systematic variation added to certain classes, which may have been a result of instrument or other selection biases. This could have also been simulated. While MNIST / CIFAR / etc are also just toy datasets that maybe are far removed from the experimental setting that the authors are envisioning, they contain high-dimensional structure on which successful pivoting would greatly strengthen the paper. Another point of concern is I don't get a clear understanding of how to choose the nuisance prior and how those choices affect results. Finally I think a big application could be an agent / environment setting. Have the authors thought about this and can they comment? L24: it would be good to have some citations here

Reviewer 3



The paper considers the problem of learning the parameters $\theta_f$ of a model $f(X;\theta_f)$ (for example a regression or a classification model) such that $f(X;\theta_f)$ is a pivotal quantity with respect to some nuisance variables $Z$. A pivotal model $f(X;\theta_f)$ is insensitive to the variations of $Z$. In other words, the objective is to train classifiers (or regressors) that are both optimal with respect to the parameters $\theta_f$ and robust with respect to the variables $Z$. The authors propose a framework that relies on the use of adversarial networks for this purpose. In particular, the output of the model $f(X;\theta_f)$ will be the input of an adversarial model that has an antagonistic objective with respect to model $f(X;\theta_f)$. It is shown that the minimax solution of this two-player game will be a model that is both optimal and pivotal (assuming that such a model exists). I think that this application of adversarial networks is novel and interesting for the community of NIPS. As far as I can understand, the theoretical results of Section 4 are correct, and the value of the approach is demonstrated by an interesting case study.